

# Whole genome sequencing analysis identifies recurrent structural alterations in esophageal squamous cell carcinoma

Munmee Dutta[1,2], Hidewaki Nakagawa[3], Hiroaki Kato[4], Kazuhiro Maejima[3], Shota Sasagawa[3], Kaoru Nakano[3], Aya Sasaki-Oku[3], Akihiro Fujimoto[5], Raúl Nicolás Mateos[1,2], Ashwini Patil[2], Hiroko Tanaka[2], Satoru Miyano[2,6], Takushi Yasuda[4], Kenta Nakai[1,2] and Masashi Fujita[3]

[1] Department of Computational Biology and Medical Sciences, Graduate school of Frontier Sciences, The University of Tokyo, Chiba, Japan
[2] Human Genome Center, The Institute of Medical Science, The University of Tokyo, Tokyo, Japan
[3] Laboratory for Cancer Genomics, RIKEN Center for Integrative Medical Sciences, Yokohama, Japan
[4] Department of Surgery, Faculty of Medicine, Kindai University, Osaka, Japan
[5] Department of Drug Discovery Medicine, Kyoto University Graduate School of Medicine, Kyoto, Japan
[6] Health Intelligence Center, The Institute of Medical Science, The University of Tokyo, Tokyo, Japan

Corresponding authors
Hidewaki Nakagawa,
hidewaki@riken.jp
Masashi Fujita, m-fujita@riken.jp

## ABSTRACT

Esophageal squamous cell carcinoma (ESCC) is the predominant type of esophageal cancer in the Asian region, including Japan. A previous study reported mutational landscape of Japanese ESCCs by using exome sequencing. However, somatic structural alterations were yet to be explored. To provide a comprehensive mutational landscape, we performed whole genome sequencing (WGS) analysis of biopsy specimens from 20 ESCC patients in a Japanese population. WGS analysis identified non-silent coding mutations of *TP53, ZNF750* and *FAT1* in ESCC. We detected six mutational signatures in ESCC, one of which showed significant association with smoking status. Recurrent structural variations, many of which were chromosomal deletions, affected genes such as *LRP1B, TTC28, CSMD1, PDE4D, SDK1* and *WWOX* in 25%–30% of tumors. Somatic copy number amplifications at *11q13.3* (*CCND1*), *3q26.33* (*TP63/SOX2*), and *8p11.23* (*FGFR1*) and deletions at *9p21.3* (*CDKN2A*) were identified. Overall, these multi-dimensional view of genomic alterations improve the understanding of the ESCC development at molecular level and provides future prognosis and therapeutic implications for ESCC in Japan.

## INTRODUCTION

Esophageal cancer is the eighth most aggressive cancer type and sixth most common cause of cancer-related death worldwide (*Zhang et al., 2015*). Esophageal cancer has two major subtypes: esophageal squamous cell carcinoma (ESCC) and esophageal adenocarcinoma (EAC). The incidence rate of ESCC is high in the Asian regions including Japan, China

and India (*Sawada et al., 2016*; *Chattopadhyay et al., 2010*). On the other hand, EAC predominates in the Western countries. Alcohol drinking and tobacco smoking are the two main risk factors of ESCC development (*Zhang et al., 2015*). Additionally, micronutrient deficiency and genetic variants that harm the activity of alcohol-metabolizing enzymes also promote ESCC (*Chang et al., 2017*). Despite the advancement of the diagnostic techniques and treatment of ESCC, the survival rate is still poor.

In Japan, ESCC is most conventionally treated by the standard neo-adjuvant chemotherapy followed by surgical resection. Preoperative chemotherapy with cisplatin and fluorouracil is considered as standard treatment option for patients in advanced stage of ESCC (*Baba et al., 2014*; *Yoshida et al., 2018*). Response rate to this standard therapy is moderate (35–40%) (*Yoshida et al., 2018*). The varying response among patients might be partly attributed to the genetic heterogeneity of tumors.

Although ESCC is common in China and Japan, ESCC of both the countries have some common as well as different characteristics. Smoking and alcohol drinking are attributed as concerning risk factors for ESCC of both the countries. In Japan, ESCC is the tenth most common cancer type while it is the fourth most frequent cancer type in China (*Sawada et al., 2016*). ESCC incidence and mortality rates are higher in China than in Japan (*Yingsong et al., 2013*). The incidence of esophageal cancer is higher in males than in females, for example, 16,241 male cases and 3,778 female cases were found in Japan in 2018 (source of data: World Health Organization [WHO] Global Cancer Observatory database) (*Yingsong et al., 2013*).

Recently, several studies in China and Japan characterized somatic mutations in ESCC using whole-exome sequencing (WES). These WES studies reported frequent mutations of *TP53*, *CDKN2A*, *NOTCH1*, *RB1*, *ERBB2* and *NFE2L2* in ESCC (*Sawada et al., 2016*; *Song et al., 2014*; *Qin et al., 2016*). While somatic mutations are important, structural variations (SVs) and somatic copy number alterations (CNAs) would also affect the development of ESCC. SVs have the capability to rearrange the large genomic alterations which impact in the treatment and prediction of its consequences in patients. However, systematic characterization of somatic mutations, mutational signatures, SVs and CNAs together have not been reported in Japanese ESCC at the whole-genome level.

In this study, we investigated 20 ESCC samples in a Japanese population using WGS. We comprehensively analyzed and showed somatic mutations in important genes, mutational signatures and their association with clinical features. Our SV and CNA analysis also identified potential target genes and regions in ESCC. The characterization of the mutational landscape in Japanese ESCC will guide our understanding of the disease in a better way and provide potential targets for the precision treatment and therapeutic prevention.

## MATERIALS & METHODS

### Clinical samples

Tumor and normal samples were obtained from 20 patients in Kindai University hospital, Osaka, Japan. All the patients agreed to participate in the study and provided written informed consent following ICGC guidelines. The study was approved by the Institutional

**Table 1   Summary of the clinical information of the ESCC samples used in this study.**

| | | |
|---|---|---|
| Sex | Male | 13 |
| | Female | 7 |
| Histology | Squamous cell carcinoma | 19 |
| | Basaloid squamous cell carcinoma | 1 |
| Tumor location | Cervical esophagus | 1 |
| | Upper thoracic esophagus | 2 |
| | Middle thoracic esophagus | 13 |
| | Lower thoracic esophagus | 3 |
| | Abdominal esophagus | 1 |
| Tumor stage | II | 2 |
| | III | 14 |
| | IV | 4 |
| Age | $\geq$60 | 18 |
| | <60 | 2 |
| Smoking status | Smoker | 14 |
| | Non-smoker | 6 |
| Alcohol drinking status | Drinker | 16 |
| | Non-drinker | 4 |
| Response to chemotherapy | Responder | 10 |
| | Non-responder | 10 |

Review Board at Kindai University Hospital and RIKEN (approval number 25-031) and Personal history of ESCC in these patients were unavailable. All except one patients (OK047) had not received any cancer treatment before the sample collection. OK047 received cisplatin-based chemotherapy before sample collection. Tumor tissues in esophagus were collected by biopsy, and histologically confirmed as ESCC. The patients were treated with neoadjuvant chemotherapy after collecting the samples. The clinico-pathological data are available in the Table 1 and Table S1.

## Whole genome sequencing

We performed WGS of the 20 pairs of matched tumor and normal samples. The tumor DNA was extracted from the ESCC samples, and normal DNA was from the lymphocytes in blood. The libraries were prepared using TruSeq Nano DNA Library Prep Kit (Illumina) following the manufacturer's protocol. Paired-end sequencing of 101- or 126-bp reads was performed using HiSeq2000/2500. The Fig. S1 shows the schematic representation of the WGS analysis pipeline performed in this study. Sequence reads were mapped to the human reference genome GRCh37 using BWA. We removed PCR duplicates using Picard tool (http://broadinstitute.github.io/picard/).

## Somatic mutation calling and mutation signature profiling

Somatic single nucleotide variations (SNVs) and short insertions/deletions (INDELs) were called as previously described (*Fujimoto et al., 2016*). Functional annotation of the

detected SNVs and INDELs was performed with Annovar (*Wang, Li & Hakonarson, 2010*). We applied dNdScv method (*Martincorena et al., 2017*) to search for genes with significant recurrent mutations ($q$-value < 0.05). We further summarized and visualized the annotated variants using the MAFtools package (*Mayakonda et al., 2018*) of the R software (https://www.r-project.org/). For the detection of mutational signatures in 20 ESCC, we used the SignatureAnalyzer (https://software.broadinstitute.org/cancer/cga/msp). Identified mutational signatures were compared with the COSMIC mutational signatures (version 2) using the cosine similarity scoring.

## Structural variation calling

Somatic SVs were called by merging calls of two software: in-house pipeline (*Fujimoto et al., 2016*) and Genomon2 structural variation (SV) detection tool (https://github.com/Genomon-Project/GenomonSV). In the Genomon2 SV detection, The SVs were called using minimum junction number 2, maximum control variant read pair 10 and minimum overhang size 50. The SVs were then filtered using parameters minimum allele frequency 0.07, maximum control variant read pair 1, control depth threshold 10 and minimum overhang size 100. The inversion size threshold set to 1,000 and the simple repeats were removed. Here, SVs were categorized into four classes based on the mapping information for a read pair. The four classes are intrachromosomal deletion, inversion and tandem duplication, and interchromosomal translocation, respectively.

The breakage-fusion-bridge (BFB) events were identified based on the information of fold-back inversion and loss of telomere (*Hermetz et al., 2014*). We implement the following criteria in order to infer BFB: (i) Inversion is single inversion (either forward or reverse) i.e., without any reciprocal partner, (ii) Inversions must have copy number change versus the adjacent position, and (iii) The two ends of the fold-back inversion must be separated by <20 kb. To detect kataegis, we used Maftools (*Mayakonda et al., 2018*) package, which is defined by mutation clusters of six or more consecutive mutations localized in a small region with an average inter-variant distance of less than or equal to 1 kb. Chromothripsis was inferred in ESCC samples based on the criteria provided by a previous study (*Korbel & Campbell, 2013*). Chromothripsis was identified in samples which show clustering of SV breakpoints, usually more than 10 breakpoints within 50 kb, with regular oscillation of copy number states.

## Copy number alteration calling

Somatic CNAs were called by analyzing read depth of matched tumor and normal using the Varscan2 software (*Koboldt et al., 2012*). The thresholds used to call the CNAs using CopyNumber function were $p$-value 0.001, minimum segment size 100 and maximum segment size 1000. The output of the above step was filtered using the copyCaller function. The raw CNAs were segmented by the circular binary segmentation method implemented in the R package DNAcopy. The GISTIC2.0 algorithm was used to identify the significant recurrent copy number amplified and deleted regions (*Mermel et al., 2011*).

### Identification of druggable genes

To identify the druggable genes across the ESCC samples, we used online gene-drug interaction database (DGIdb) (*Griffith et al., 2013*). We used the genetically altered genes as target data in order to determine the druggability of the genes. The target genes were detected by different method such as SNV/INDEL analysis, SV analysis and CNA analysis in this study. The database provides two options to examine gene-drug interactions either by gene or by drug names. In this case, we identified gene-drug interactions by providing the gene names. The genes that have at least one interaction with drug target was considered as druggable gene.

## RESULTS

### Whole genome sequencing of ESCC samples

To identify the mutational events and driver genes that contributing to the development of ESCC in Japanese population, we performed WGS of 20 pairs of tumor and matched blood samples. The samples were collected by biopsy from the individuals before neo-adjuvant chemotherapy (except OK047). Among the 20 samples, 10 samples responded well to the therapy while rest 10 samples showed poor response. The average genome coverage was 43. $4\times$ for the tumor samples and 34. $3\times$ for blood samples, after removal of polymerase chain reaction (PCR) duplicates. The WGS data was computationally analyzed to call somatic alterations of the following types: single nucleotide variations (SNVs), small insertions and deletions (INDELs), structural variations (SVs), and copy number alterations (CNAs). In all, WGS analysis identified 104,534 somatic SNVs, 10,523 somatic INDELs, and 2,641 somatic SVs in the 20 tumors (Table S2). In addition, WGS analysis detected 10 significant copy number altered regions in ESCC.

### Recurrent coding mutations in ESCC

We investigated the somatic SNVs, short INDELs in the protein-coding regions and their splice sites in the 20 ESCCs. Our WGS analysis identified recurrently mutated genes, including previously known esophageal cancer associated oncogenes and tumor-suppressor genes. We evaluated the somatic alterations by dNdSCV (*Martincorena et al., 2017*) method in order to find out the significantly recurring mutations across the ESCC genomes. We identified significant mutations in *TP53* and *ZNF750* genes in ESCC consistent with findings by previous studies (Fig. 1 and Fig. S2). *TP53* mutations were most frequent and found in 55% of the samples followed by *ZNF750 (15%)*. In addition, other genes that recurrently mutated in ESCC include *FAT1 (10%), PTCH1 (10%), EP300 (5%), FAT2 (5%), FBXW7 (5%), KMT2D (5%), NFE2L2 (5%), NOTCH1 (5%), PIK3CA (5%), RB1 (5%), RIPK4 (5%)* and *TP63 (5%)*. These genes were previously reported in ESCC by different studies (*Zhang et al., 2015*; *Sawada et al., 2016*; *Chang et al., 2017*; *Qin et al., 2016*; *Li et al., 2018*; *TCGA, 2017*).

### The mutational signatures of ESCC

In order to understand mutational mechanisms of ESCC in Japan, we analyzed the mutational signatures of the 20 ESCC tumors. A Bayesian variant of the non-negative

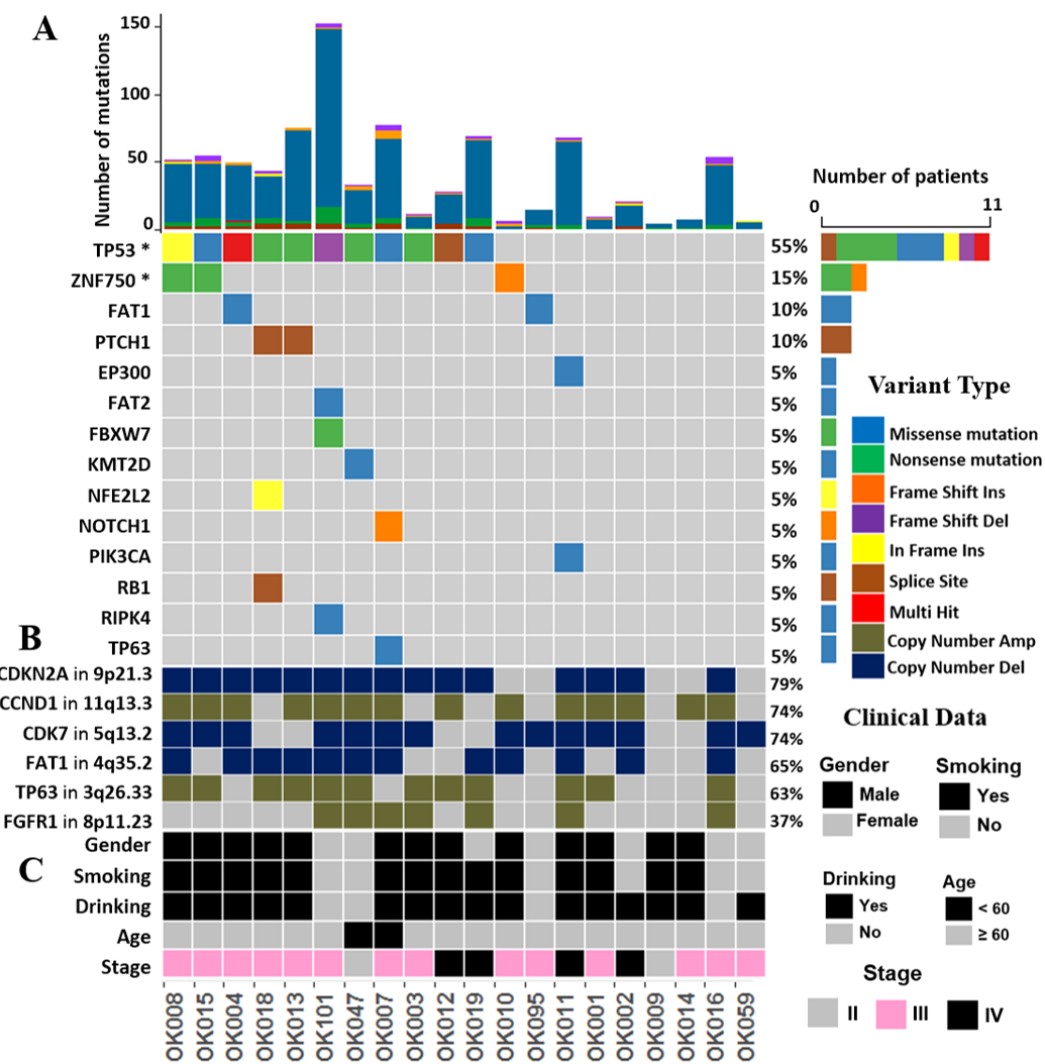

**Figure 1 The landscape of somatic alternations in ESCCs from 20 Japanese patients.** (A) Potential driver mutations by SNVs/INDELs across the 20 ESCC patients with different mutation types coded by different colors. Two genes marked with an asterisk are significantly mutated genes ($q < 0.05$) detected by the dNdScv method. The other 12 genes are those recurrently mutated in previous ESCC studies. (A) shows the number of mutations in all the 20 ESCC cases. The number and type of mutations for each mutated gene is also shown. Mutation types are labelled on the legend. (B) The recurrent copy number amplified and deleted regions with the important cancer-related genes detected in ESCC patients. The legend shows frequency across the ESCC patients. (C) The clinical features such as gender, smoking, alcohol drinking status, age and the tumor stages of the ESCC patients.

matrix-factorization method was applied to trinucleotide substitution patterns and extracted six mutational signatures (Figs. 2A–2F). The mutational signatures in ESCC were then compared with the signatures of the Catalogue of Somatic Mutations in Cancer (COSMIC) database *Alexandrov et al., 2013* (Table S3).

The identified Signature W1 was highly similar to the COSMIC signature 18 (cosine similarity 0.933), but the biological aetiology of this signature is not known. Signature

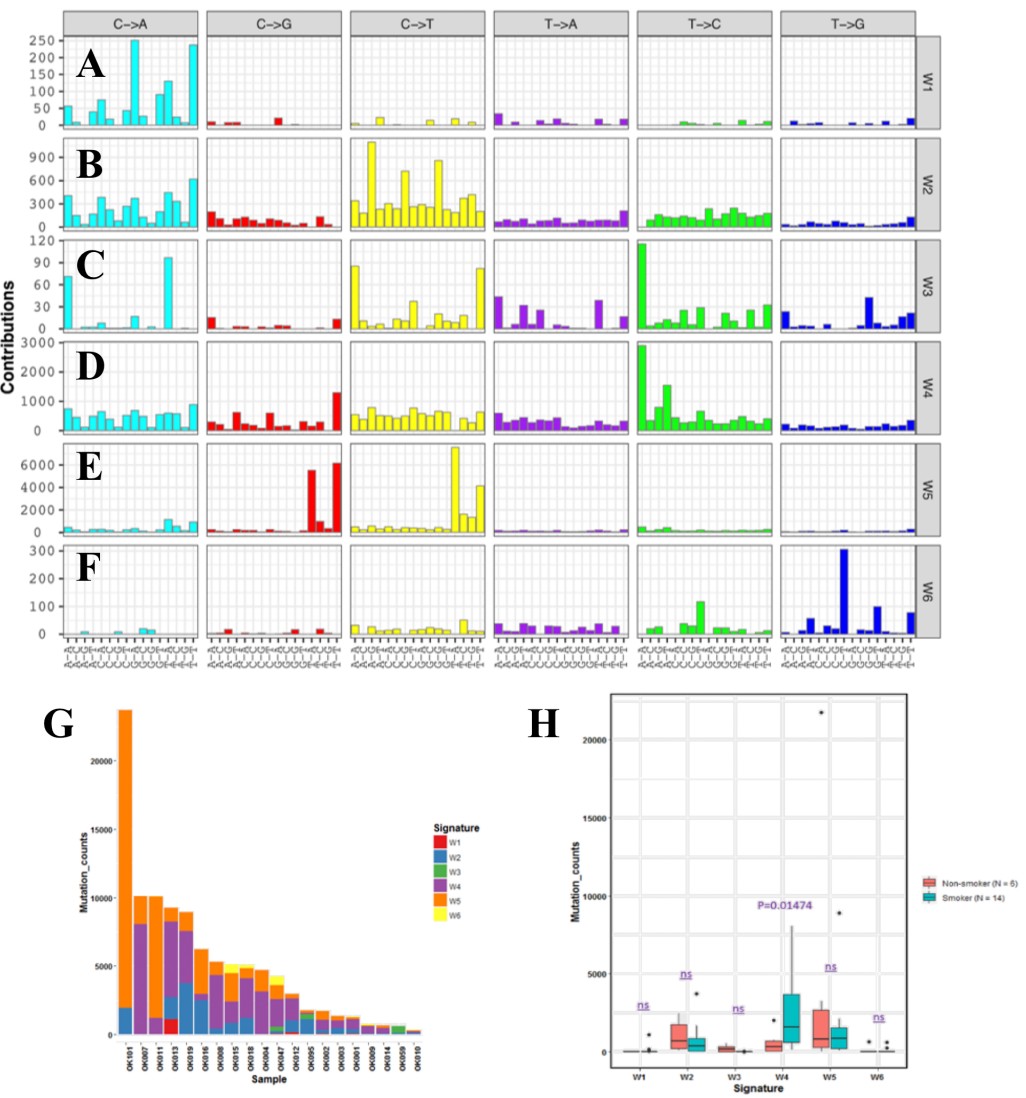

**Figure 2** **Mutational signatures of 20 whole genome of ESCC.** (A–F) Characterization of six mutational signatures identified across the ESCC genomes. Patterns of substitutions for each signatures W1–W6 in ESCC. The mutational signatures are presented according to the 96 substitution classifications defined by the substitution class and sequence context immediately 3' and 5' to the mutated base. We used the SignatureAnalyzer method to determine the six distinct mutational signatures (SNVs) out of the 20 ESCC samples. (A) Signature W1. (B) Signature W2. (C) Signature W3. (D) Signature W4. (E) Signature W5. (F) Signature W6. (G) Mutation burden and contribution of the six mutational signatures (W1-W6) across the ESCC genomes. (H) Boxplot showing the association of mutational signatures and smoking habit of patients with ESCC. Mutational signature W4 displays significant association ($p$-value $= 0.01474$) with smoking status of ESCC patients. $P$-value was calculated using Wilcoxon rank sum test. ns: $P > 0.05$.

W2 was characterized by C>A and C>T mutations and found in almost 85% of the ESCC patients (Fig. 2G). Signature W2 was similar to the COSMIC signature 1 (cosine similarity 0.850), which is an age-dependent signature. Signature W4 mainly represented by T>C mutation, was similar to COSMIC signature 5 and 16 (cosine similarity 0.868 and 0.9,

respectively. Signature W5 displayed high similarity with COSMIC signature 2 and 13 (cosine similarity 0.811 and 0.835, respectively), which was characterized by C>T and C>G mutations. COSMIC signatures 2 and 13 were assigned to the hyperactivity of the APOBEC family enzyme, cytidine deaminases (*Alexandrov et al., 2013*). One of our samples, OK101, was basaloid squamous cell carcinoma of esophagus, a rare type of malignancy, and a hypermutator in our cohort (Fig. 2G). Somatic SNVs of this sample were dominated by the Signature W5. Signature W6 was characterized by CpTpT-to-CpGpT mutations and highly similar to COSMIC signature 17 (cosine similarity 0.945). COSMIC signature 17 is often found in esophagus and stomach cancer, but its aetiology is unknown. Signature W3 which was defined by C>A, C>T and T>C mutations appeared to have low similarity with any of the COSMIC signatures (all cosine similarity <0.7).

Since, previous studies did not find signatures associated with smoking alone in ESCC despite smoking being a major risk factor of this cancer (*Sawada et al., 2016*; *Zhang et al., 2015*). We examined the association of the mutation signatures with clinical features in ESCC. We found signature W4 was significantly elevated in the smoking patients compared to non-smoking patients ($P = 0.01474$, $t$-test) (Fig. 2H). Mutations of W4 may be caused by the carcinogenic chemical in tobacco smoke. Signature W4 shared similarity with COSMIC signature 5 and 16. Smoking associated mutational signatures were found higher in the squamous cell carcinomas of lung and head & neck (*Wang et al., 2019*). However, there was no difference in total mutation burden between the smoker and non-smoker group ($P = 0.7$, $t$-test) unlike liver cancer. In liver cancer patients, similar signature showed higher mutation rate in smoker group than the nonsmokers (*Alexandrov et al., 2016*). The other signatures showed no association with smoking, gender, response to chemotherapy and alcohol drinking status ($P > 0.05$).

## Structural variations in ESCC

WGS analysis detected a total of 2,641 SVs in the 20 ESCC samples with an average of 132 SVs per tumor. The number of SVs varied, ranging from 0 to 515 across the 20 ESCC cases which shows the heterogeneous nature of tumor genome (Fig. S3). In particular, compared to other ESCC cases OK007 and OK008 had high number of SVs which affected most of their chromosomes, indicating genomic instability in these samples (Figs. 3A–3L and Fig. S4). The deletions were the most abundant type of SVs across the samples (Fig. 3M). We found 1,090 (41.27%) deletions followed by 793 (30.03%) inversions, 391 (14.80%) translocations and 367 (13.90%) tandem duplications, respectively.

We found numerous cancer associated genes affected by SVs in ESCC (Fig. 3N and Table S4). Consistently with a report on Chinese ESCC (*Chang et al., 2017*), *LRP1B* (30%) and *TTC28* (30%) were the most commonly affected genes in Japanese ESCC. The tumor suppressor gene *LRP1B* was mostly affected by deletions, whereas *TTC28* was by interchromosomal translocations. We were also able to identify *SDK1*, a novel gene, affected by SVs in 25% of the ESCC samples. *SDK1* is a cell adhesion molecule that plays an active role in cancer development. Somatic mutations in this gene was reported in adrenocortical carcinoma (*Juhlin et al., 2015*). Recurrent SVs in *CSMD1, WWOX, ERC1, PDE4D* and *SHANK2* were also identified in five tumor samples. The CUB and Sushi multiple domains
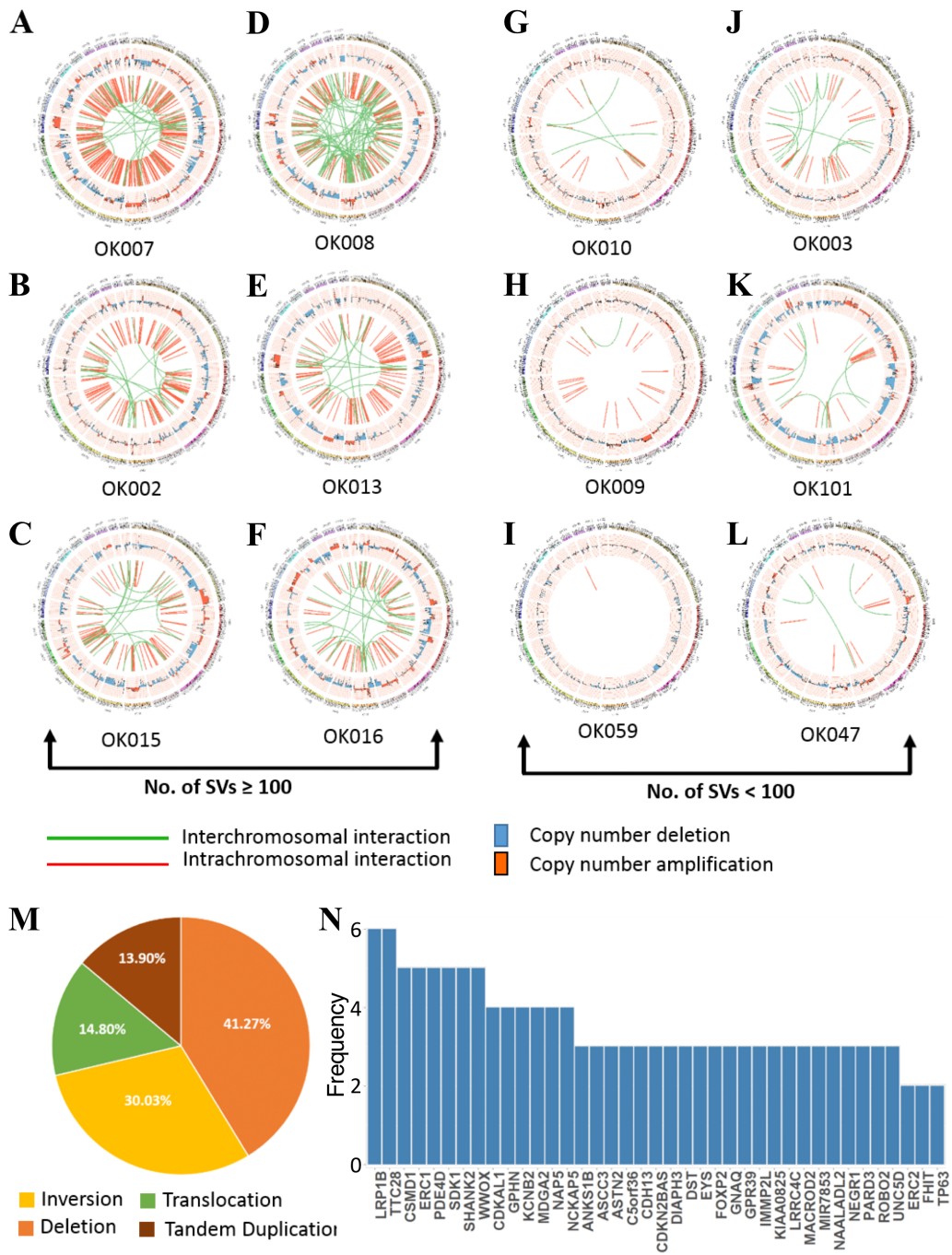

**Figure 3** **Somatic structural variations in ESCC.** (A–L) Circos plots of SVs and somatic CNAs in 20 ESCC genome. The inner ring represents the SVs: red for intrachromosomal rearrangements, and green for interchromosomal rearrangements. The second ring next to SVs displays the CNAs: red for amplifications and blue for deletions. The outer ring shows the chromosome ideogram. (A–F) ESCCs with 100 or more SVs. (G–L) ESCCs with <100 SVs. (M) Different categories of SVs for instance inversion, translocation, deletion and tandem duplication, and their frequencies in the 20 ESCC genomes. (N) Top genes recurrently affected by SVs across the ESCC samples. Genes that were affected in two or more patients are presented.

1 (*CSMD1*) is a tumor suppressor gene reported to be associated with poor prognosis in many cancer types including breast cancer, gastric cancer, head and neck squamous cell carcinoma (HNSCC), and hepatocellular carcinoma (*Deng et al., 2012*; *Zhang et al., 2019*; *Jung et al., 2018*). Deletion of *WWOX,* a tumor suppressor gene, is frequent in esophageal adenocarcinoma (32%) and stomach adenocarcinoma (30.2%) and also observed in other human cancer types such as colon adenocarcinoma, bladder urothelial carcinoma and lung adenocarcinoma (*Hussain et al., 2019*). Structural rearrangements of *ERC1* was previously reported in Chinese ESCC (*Chang et al., 2017*). *ERC1* was also found as a prognostic biomarker in HNSCC (*Szczepanski et al., 2013*). Previously, a genome wide association study in Chinese Han population identified that SNP rs10052657 in *PDE4D* on 5q11 was associated with ESCC risk (*Wu et al., 2011*). Homozygous deletion of *PDE4D* was also identified in breast, lung and gastric cancers which established it as a tumor-promoting gene (*Lin et al., 2013a*; *Lin et al., 2013b*).

Breakage-Fusion-Bridge (BFB) is a mechanism supported by previous studies in cancer (*Cheng et al., 2016*; *Yang et al., 2017*). The BFB event is characterized by a special type of structural rearrangements called 'fold-back' inversion. Fold-back inversion can be defined as somatic structural variants with single inverted breakpoints exhibiting copy-number changes (*Campbell et al., 2010*). We implemented these information to identify BFB in each ESCC genome. In total, we detected 101 fold-back inversions across the 20 ESCCs, of which chromosome 11 appeared to have highest fold-back inversions (30) (Fig. S5A). BFB event was present in total 14 ESCC cases (70%) in this study. Moreover, fold-back inversions were observed on chromosome 11 around amplification of *CCND1* locus (69455873-69469242) in eight ESCC cases (Fig. S5B). Notably, our CNA analysis identified amplification of *CCND1* in fourteen patients, and 57% of those amplifications (8/14) were caused as a result of BFB. In addition, oncogenes such as *FGFR1* (1/20) (Fig. S5C) and *EGFR* (1/20) were also detected in the amplified regions which were affected by BFB event. In all, this analysis presented an important insight of BFB in ESCC, and targeting the amplified oncogenes in therapies will benefit the ESCC patients in future.

Kataegis loci, which are localized hyper-mutation clusters, were identified in 30% of the ESCC patients (6/20) (Fig. S6A and Table S5). In total 11 kataegis loci were detected in six cases, of which four kataegis had SVs in their close vicinity. Previous studies reported kataegis in ESCC (*Chang et al., 2017*; *Cheng et al., 2016*), and in breast cancer it was observed in more than 50% of the cases (*Nik-Zainal et al., 2012*; *D'Antonio et al., 2016*). Furthermore, chromothripsis (*Korbel & Campbell, 2013*), a phenomenon that affects chromosomes with more than ten structural rearrangements with regular oscillation of copy number changes, was found in one ESCC patient (OK008) (Fig. S6B) on chromosome 12 and 14. Chromothripsis lead structural rearrangements have been reported in many cancers usually in low frequency, however, more than 40% chromothripsis were found in glioblastomas and lung adenocarcinomas (*Korbel & Campbell, 2013*; *Cortes-Ciriano et al., 2020*).

## Somatic copy number alterations in ESCC

We called the somatic CNAs from the whole genome of the 20 pairs of matched tumor and normal samples to investigate CNAs in ESCC. GISTIC2.0 was then used to identify the recurrently amplified and deleted regions (Mermel et al., 2011). We identified four frequent amplified regions (*3q26.33, 8p11.23, 11q13.3* and *14q21.1*) and six deleted regions (*1q21.1, 4q35.2, 5q13.2, 9p21.3, 10p12.33* and *21p11.2*) (Figs. 4A and 4B, Tables S6 and S7).

The copy number amplification of *3q26.33* was found in 63% of the samples and included important caner driver genes *PIK3CA, SOX2, FGF12* and *TP63*. Notably, SVs in *TP63* was also found in 15% of the ESCC samples in this study. It was observed that *11q13.3* was the most frequently amplified region (74%) in our dataset, consistently with the observation in Chinese ESCC (Ying et al., 2012). *11q13.3* gain harbored many important genes such as *CCND1, FGF3, FGF4 and FGF19* which established this region as a prominent target in ESCC. Amplification of *8p11.23* involved *FGFR1*, which plays an active role in cell growth and differentiation. Earlier studies reported *FGFR1* as a potential drug target in many human cancers (Von Loga et al., 2015; Chang et al., 2014; Lin et al., 2014).

Copy number deletion of *9p21.3* was found in 79% of the samples and it was the most common deletion in the ESCC. This region contains *CDKN2A*, an essential regulator of cell cycle (Fig. 4C). *10p12.33* was deleted in 79% of the samples and harbored gene *MRC1*, a M2 macrophage antigen known to be associated with tumor development, invasion, metastasis and angiogenesis (Weber et al., 2016; Fang et al., 2017). Deletion of *4q35.2* was found in 63% of the samples. This region includes the tumor suppressor *FAT1*, which was recurrently deleted in colorectal cancers, glioblastoma and HNSCC (Morris et al., 2013). Notably, recurrent SNVs/INDELs in *FAT1* was observed in our study as well. Overall, this study detected many copy number altered peaks and important ESCC-associated driver genes such as *FGFR1, PIK3CA, CCND1, CDKN2A* and *MRC1*, which have the potential to be used as therapeutic targets in future (Du et al., 2017; Lin et al., 2014; Padhi et al., 2017; Weber et al., 2016; Weber et al., 2014).

## Assessment of druggable genes in ESCC

In order to examine the druggability of the genes identified by the CNA analysis, we analyzed the genes with the drug-gene interaction database (Griffith et al., 2013; Cotto et al., 2018). We found that 67 genes out of the 442 genes had at least one drug target that interact with it (Table S8). *HTR1A, PIK3CA, FGFR1, ADRB3, HPIK3R1, MTNR1A, CDKN2A, CCND1, CDK7,* and *ANO1* were the top druggable genes that showed large interactions with drugs. Notably, amplification of *CCND1* (74%), *PIK3CA* (63%), *FGFR1* (37%) and deletion of *CDKN2A* (79%) and *CDK7* (74%) genes were observed in a considerable amount of samples in our analysis. In all, CNA analysis was able to identify potentially druggable genes including *CCND1, FGFR1, PIK3CA* and *CDKN2A* which might be used for the treatment of ESCC in future.

The druggability of genes detected by our SNV analysis was also examined. We identified 212 genes out of the 1,111 genes that have at least one interaction with drug (Table S9). We identified *PIK3CA, TP53, BRAF, NOTCH1, FGFR2, F11, MTOR, LRP2,* and *RB1* were the most prominent druggable genes with large number of drug-interactions. On the
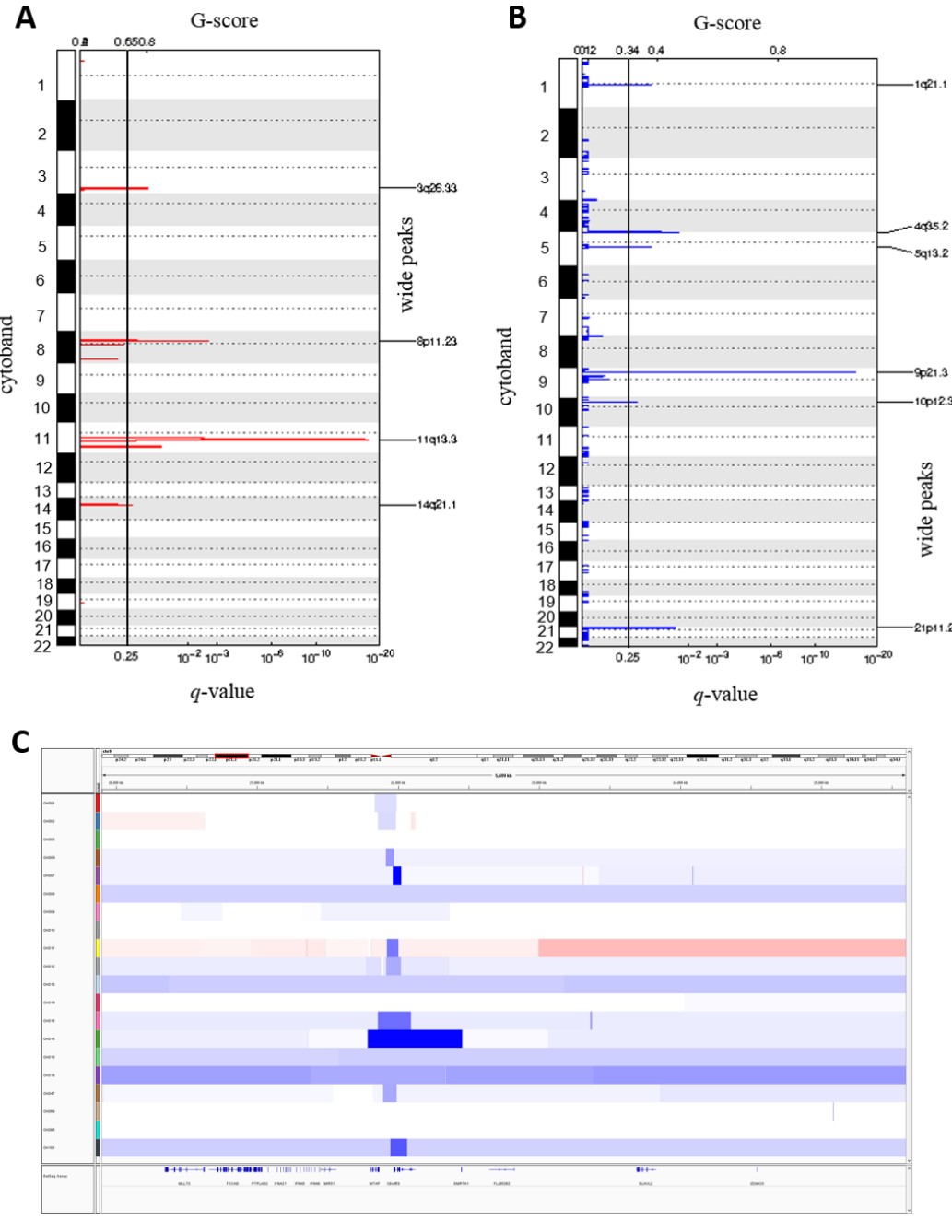

**Figure 4  Somatic copy number alterations in specific regions detected by GISTIC 2.0 in ESCC.** Significantly observed regions of recurrent amplifications (A) and deletions (B) across samples are shown. Numbers in the left bar in both (A) and (B) refer to the chromosome number. GISTIC scores are presented on top and, *q*-values (*x*-axis) indicating the false discovery rate at each locus are shown on a log scale in both (A) and (B). (C) Common deletions are shown at 9p21.3 region across the ESCC samples by Integrative Genomic Viewer (IGV).

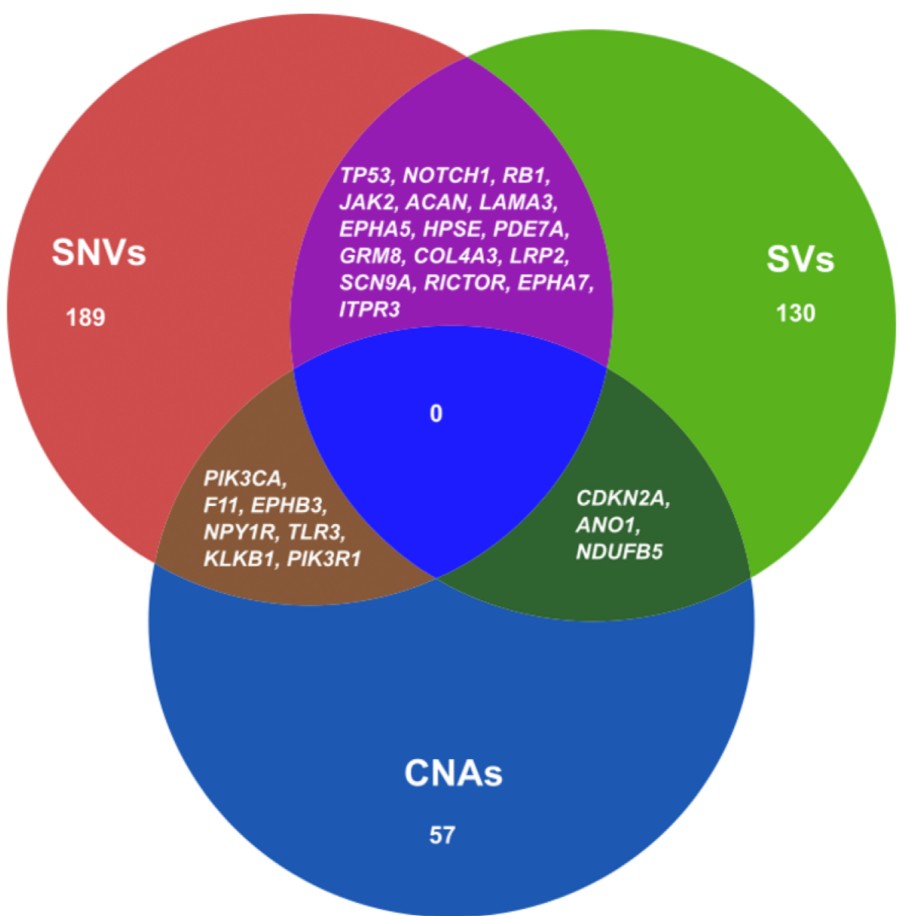

**Figure 5** **Venn diagram representing the druggable genes altered by different mutation events in ESCC.** The figure shows the common druggable genes, genes having interaction with at least one drug target. These genes were affected by SNVs, SVs and CNAs in ESCC.

other hand, 149 genes out of 958 have the druggable property which were identified by SV analysis here (Table S10). *LRP1B, PDE4D, WWOX, GPHN, KCNB2, FHIT, CDKN2A, TP53* and *BRCA2* were the important druggable genes determined with large number of interactions with drugs. *LRP1B* (30%), *PDE4D* (25%), *WWOX* (25%), *GPHN* (20%) and *KCNB2* (20%) were frequently affected by SVs in ESCC.

We also combined the analysis of druggable genes identified by all the mutational events SNVs, SVs and CNAs in ESCC. It was observed that some of the genes such as *TP53, NOTCH1, RB1, JAK2, ACAN* and *SCN9A* were commonly altered by both SNVs and SVs in ESCC (Fig. 5 and Table S11). Genes, for example, *PIK3CA, F11, EPHB3* and *TLR3* were commonly altered by SNVs and CNAs. We also detected some genes such as *CDKN2A, ANO1* and *NDUFB5* were altered by both SVs and CNAs. However, we found no druggable gene that was altered commonly by all the three mutation patterns in our analysis.

## DISCUSSION

In this study, we performed a comprehensive whole genome sequencing analysis in order to explore mutational landscape in ESCC with Japanese origin. Previously, several studies reported mutations in ESCC mostly using whole exome data, which is limited to coding mutations. Here, we provided a complete whole-genome level analysis of mutational signatures, SNVs, SVs and CNAs in ESCC. Consistent with the previous studies, this study also confirmed the frequent mutation of *TP53, ZNF750, FAT1, PTCH1, EP300, FAT2, FBXW7, KMT2D, NFE2L2, NOTCH1, PIK3CA, RB1, RIPK4* and *TP63* in ESCC patients (*Chang et al., 2017*; *Sawada et al., 2016*; *Qin et al., 2016*). Recurrent mutations in most of these genes were previously reported in HNSCC (*Lin et al., 2018*; *Hazawa et al., 2017*; *TCGA, 2015*) and lung squamous cell carcinoma (*Choi et al., 2017*; *Li et al., 2015*).

By analyzing mutational signatures we showed distinct signature profiles in ESCC and their correlation with environmental risk factor and clinical features. Importantly, among the identified six mutational signatures, significant association was shown between signature W4 and smoking status of ESCC patients. This signature W4 showed higher similarity with COSMIC signature 5 and 16 (cosine similarity 0.868 and 0.9, respectively). A WGS study showed association of COSMIC signature 16 with alcohol drinking and smoking status in ESCC patients of Chinese origin (*Chang et al., 2017*). Smoking tobacco increases the chances of cancer development. Tobacco smoke is composed of many carcinogens which are capable enough to disrupt the DNA. Tobacco smoking has been linked to many type of cancers such as lung cancer, liver cancer, esophageal cancer and gastric cancer (*Alexandrov et al., 2016*). COSMIC Signature 5, which is also characterized by C>T and T>C mutations, was reported in smoking associated lung cancer. Smoking associated signature was more commonly found in male patients (65%) with smoking history than female, and 62% of those patients were non-responder to chemotherapy. In contrast, the female patients (30%) who showed this signature were mostly non-smoker (86%). However, the female patients with nonsmoking history who showed signature W4, 84% of them responded well to the chemotherapy.

A recent WGS analysis on UK patients suggested C>A/T as a distinct mutational pattern with evidence of ageing in EAC (*Secrier et al., 2016*). We confirmed that C>T and T>C substitutions were also dominant type mutations with age and smoking imprint in ESCC as well. Further, it was previously described that these mutation patterns characterize the smoking and alcohol drinking signatures by *Alexandrov et al. (2016)*. On the other hand, mutational signatures identified here showed high similarity not only with COSMIC signatures associated with alcohol drinking and smoking but also with age and APOBEC signature activity related signatures. APOBEC family enzymes alter cytidine to uracil through the process of deamination within DNA which leads to mutation clusters in different cancers (*Harris, Petersen-Mahrt & Neuberger, 2002*). Previous studies reported the association of *PIK3CA* mutations with APOBEC signature in Chinese and Japanese ESCC (*Chang et al., 2017*; *Sawada et al., 2016*; *Zhang et al., 2015*). In this study, we found that APOBEC signature is positively associated (*P = 0.00003969*, Wilcoxon rank sum test) with amplification of *PIK3CA*. APOBEC signature was observed in all the ESCC tumor

samples (100%). APOBEC mediated genomic damages may play a major role in ESCC development.

In the present study, we analyzed SVs using WGS of 20 ESCCs which was the first time report in Japanese population. A recent analysis of WGS of 94 ESCCs from China showed *LRP1B* and *TTC28* as the most commonly affected genes by SVs (*Chang et al., 2017*). We confirmed this finding in Japanese ESCCs as well. *LRP1B* was mostly affected by deletions and *TTC28* by interchromosomal translocation in all the cases. Further, this analysis identified SVs in a novel gene *SDK1* in 5 out of the 20 ESCC tumor samples. An epigenomic study identified *SDK1* as an epigenomic driver in hepatocellular carcinoma (*Gentilini et al., 2017*). Besides, SVs were found in other genes such as *WWOX, CSMD1, ERC1, PDE4D, SHANK2,* and *TP63.* This study found two patterns of structural rearrangements across the 20 ESCC genome. In the first pattern samples were found with a few rearrangements and in the other, samples were with multiple complex rearrangements which represent the heterogeneous nature of ESCCs. Moreover, this study identified breakage-fusion-bridge, kataegis and chromothripsis in 75% of the ESCC patients (15/20). Chromothripsis, a phenomenon when multiple rearrangements affect a single or multiple chromosomes in a single event (*Rode et al., 2016*; *Korbel & Campbell, 2013*). It is believed that chromothripsis plays important role in the development of cancer. In recent years, chromothripsis has emerged as a significant biomarker in different cancers. Chromothripsis may promote cancer development either by increasing copy numbers of oncogenes or by deleting important tumor suppressors. Thus, identification of chromothripsis occurrence in tumors using strict criteria is crucial for cancer genomics. More studies are required with a comparatively larger sample size in future.

Moreover, we identified multiple significantly amplified and deleted regions in ESCC genomes. These regions harbored many genes that may be used as therapeutic targets. Amplification of *11q13.3* region, which contains *CCND1, CTTN, SHANK2* and three *FGF*-family genes, was frequently altered in patients that responded to chemotherapy. *CCND1*, a key regulator of the G1 phase of cell cycle, is an important target of chemotherapy response in HNSCC (*Feng et al., 2011*). It has been associated with poor prognosis in many solid cancers including HNSCC (*Feng et al., 2011*). Another amplification region *3q26.33* harbored important caner-causing genes such as *PIK3CA, SOX2, FGF12* and *TP63.* Alterations of *PIK3CA* which leads to dysfunction of cell cycle control, was also frequently found in colorectal cancer, HNSCC, gastric cancer and breast cancer (*Mei et al., 2016*; *DeMello et al., 2018*; *Azizi Tabesh et al., 2017*; *Du et al., 2017*). *PIK3CA* was reported to have association with drug sensitivity in many cancer types including ESCC (*Du et al., 2017*; *Yokota et al., 2018*). Furthermore, amplification of *TP63* was also commonly found in squamous cell carcinomas of lung, and neck and head which showed over-expression in these tumors (*TCGA, 2015*; *Ohnami et al., 2017*). *TP63* is a transcription factor and plays important role in tumorigenesis, apoptosis and embryogenesis (*Ohnami et al., 2017*; *Candi et al., 2014*). Deletion of *CDKN2A* gene is more frequent in Japanese than Chinese ESCC. We confirmed deletion of *CDKN2A* in *9p21.3* and *MRC1* in *10p12.33* regions as the most common deletion (79%) in our dataset. Previously several studies reported recurrent mutations and loss in *CDKN2A* in many human cancer types such as pancreatic cancer

and oral squamous cell carcinoma (*Zhen et al., 2015*; *Padhi et al., 2017*). A previous study on Chinese population also reported deletion of *CDKN2A*, but only 30% of the Chinese samples showed deletion of this gene. Our gene-drug interaction analysis was also able to detect druggable genes for example *FGFR1, PIK3CA, CCND1, CDKN2A, CDK7* and *ANO1* that have the potential to interact with at least one drug.

There were some limitations in this study. The main bottle-neck of this study was the sample size. We had 20 pairs of matched normal and tumor samples, which was not enough to find significantly mutated recurrent genes with high level of frequencies. Secondly, although the patient's chemotherapy response information was available, the small number of sample size did not allow us to associate the identified genomic alterations with chemotherapy response at a significant level. It would be important to associate the genomic alterations with chemotherapy response to select the correct patients for effective neoadjuvent chemotherapy in future.

## CONCLUSIONS

In summary, this analysis was able to identify the frequent genomic mutations, mutational signatures and significant association with environmental risk factor, druggable CNAs and genes, and structural rearrangements in ESCC. This comprehensive analysis provided insights into the ESCC development at molecular level and identified targets for the diagnosis and treatment of ESCC in future.

## ACKNOWLEDGEMENTS

The super-computing resource 'SHIROKANE' was provided by the Human Genome Center, The University of Tokyo (http://supcom.hgc.jp/).

### Funding
This work was supported by the Japan Agency for Medical Research and Development (AMED) Project for Cancer Research and Therapeutic Evolution (P-CREATE) (to Hidewaki Nakagawa) and JSPS KAKENHI Grant Number 19K09206 (to Masashi Fujita). Munmee Dutta was supported by the Japanese Government Scholarship (Monbukagakusho, MEXT) for graduate studies. There was no additional external funding received for this study. The funders had no role in study design, data collection and analysis, decision to publish, or preparation of the manuscript.

### Grant Disclosures
The following grant information was disclosed by the authors:
Japan Agency for Medical Research and Development (AMED).
Project for Cancer Research and Therapeutic Evolution (P-CREATE).
JSPS KAKENHI: 19K09206.
Japanese Government Scholarship (Monbukagakusho, MEXT).

## Competing Interests

Kenta Nakai is an Academic Editor for PeerJ.

## Author Contributions

- Munmee Dutta analyzed the data, prepared figures and/or tables, authored or reviewed drafts of the paper, and approved the final draft.
- Hidewaki Nakagawa conceived and designed the experiments, analyzed the data, authored or reviewed drafts of the paper, and approved the final draft.
- Hiroaki Kato, Kazuhiro Maejima, Kaoru Nakano and Aya Sasaki-Oku performed the experiments, authored or reviewed drafts of the paper, and approved the final draft.
- Shota Sasagawa. Akihiro Fujimoto, Raúl Nicolás Mateos, Ashwini Patil, Hiroko Tanaka, Satoru Miyano, Kenta Nakai and Masashi Fujita analyzed the data, authored or reviewed drafts of the paper, and approved the final draft.
- Takushi Yasuda conceived and designed the experiments, authored or reviewed drafts of the paper, and approved the final draft.

## Human Ethics

The following information was supplied relating to ethical approvals (i.e., approving body and any reference numbers):

The study was approved by the Institutional Review Board at RIKEN and Kindai University Hospital (25-031).

## Data Availability

Sequencing data are available in the Japanese Genotype-phenotype Archive (JGA): JGAS00000000155.

Because genetic sequences of human patients are protected, only researchers who obtain ethical approval from JGA can access the data.

The code that was used to perform the analysis and generate the figures/tables can be found at https://github.com/mfujita-riken/Munmee2020.

## Supplemental Information

Supplemental information for this article can be found online at http://dx.doi.org/10.7717/peerj.9294#supplemental-information.

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
