# Peer review of "Whole genome sequencing analysis identifies recurrent structural alterations in esophageal squamous cell carcinoma"

_PeerJ, doi:10.7717/peerj.9294_

## Round 0.1 · original submission · Major Revisions

Both reviewers have significant concerns as shown in the reports and please address them point-by-point. In the material and method part, please clarify whether these patients have received any cancer treatments before the sample collection. Are these patients first time diagnosed as ESCC?

·

Basic reporting

The study uses clear language to introduce the topic, the approach and the findings.

There were a few issues:
1. References are missing for some statements. Examples are lines 52-55 and 111 (R language).
2. The results section includes description and references to the importance/interpretations of the findings which may be appropriately placed in the discussion section (not sure what is the journal policy on this item or how strict they are on enforcing this).
3. Figure 1. Top and right annotation (bar graphs) are missing y and x-axis labels (number of mutations and number of patients). In C, binary variables can be represented by 0/1 or white/black instead of unique colors.
4. Figure S1. doesn’t show place the calling of INDELs in the pipeline. In addition, the figure can be more informative by including the tools that were used at each step.
5. Figures 3A and S3 are uninformative. They don’t clearly (obviously) show the trends described in the text referencing them.
6. Figure 4. Axes labels are not provided.
7. Figure legends: The figure legends are brief and are not self-contained. The reader would have to refer to the methods and the results to be able to read the figures. The legends for the supplementary figures are missing.
8. There is no reference to the raw data. The raw data should be deposited in the appropriate repository and the repository identifiers are listed in the manuscript (check the journal policy).

The authors provided ethical approval statement for the use of data from human subjects and no identifiable info of the subjects are in the text.

Experimental design

The authors defined their research question and documented their approach. .
Here are a few issues and suggestions:

1. The methods used to assess druggable genes are not described.
2. Line 68. The incidence rate and the risk factor association of a disease are presumably independent. This sentence presents an unclear contrast between the two facts.
3. Line 69. The authors can present direct statistics of the prevalence of the ESCC in men and women instead of relying on the mortality rate ration, which depending on how it was calculated it may or may not reflect the prevalence of the disease.
4. There is no statement on the reproducibility of the study. It is on the authors to assess the reproducibility of their study and to guide the reader to the appropriate resources (data/code/documentation) to achieve that level of reproducibility.

More issues with the methods are raised in the context of making certain claims (Validity of the findings).

Validity of the findings

The study provides useful exploration of the mutational landscape of ESCC in a specific population of a japanese city. They identified certain sequence and structural variations in the samples. However, some of the claims in the text went beyond exploration to establish correlation between the disease and the patient characteristics.

Here are two examples and why they might be problematic:

1. Specific claim such as “Enrichment of signature W4 in smokers confirms the carcinogenic role of tobacco smoking in ESCC” are harder to establish in this study for several reasons.
a. The small sample size.
b. The use of alternating logic to associate signatures with patient metadata and similarity with the COSMIC signatures. If any correlation to be established, the signatures and patient metadata from this study and the COSMIC study should be shown. But alternating between the similarity to another study and the enrichment in one study is not enough reason to establish correlations.
c. Other variables (confounders) such as gender, age or drinking were not ruled out (or not shown) as an explanation of the correlations. For example, if most smokers were males or over 60 yo, it is hard to see why maleness or old age is the reason for the apparent correlation.
d. The use of t-test and statistical significance (p-value) is unfortunate. First, the two groups are of unequal sizes (n = 14 and 5). Second, the within group variances of the two groups (although are not explicitly mentioned) look very different from the boxplot (Figure 2C).
e. The claim is contradicted in the discussion (line 312). It says that the signature W4 is only common in male patients, and female patients who had it were non-smokers and responded to chemo unlike their male counterparts.

2. Similarly, the authors claim that certain structural variants are mostly common in male “patients with smoking and drinking (line 234, table S5)”, while others are common in patients “irrespective of the same characteristics (line 236, table S5).” The latter being a no claim is hard to justify. The problem with the former is that most of the patients in the study are males and with history of drinking and smoking. If possible, the authors should support the claim with proper visualization and testing.

Reviewer 2 ·

Basic reporting

In this study Munmee Dutta et al provide a genomic map of Structural variations, CNV and somatic point mutations in ESCC using 20 Whole Genome Sequencing samples. This is a unique dataset in the Japanese population and the authors have done a commendable job in identifying the key driver Structural Variants and Copy Number events. However, the following caveats need to be addressed and this would significantly improve the findings in the manuscript.

Experimental design

The 20 Whole genome cases will be a good resource for the ESCC genomic community.

Validity of the findings

Major Comments
1) A new signature associated with smoking has been reported. Alexandrov et al have shown a smoking signature in Lung Cancer. They have further validated this signature in-vitro by exposing cell-lines to Benzopyrene, which has also resulted in the same signature. Given the vast amount of literature available on the smoking signature, Munmee at el should clarify the cosine similarity of the smoking signature. Secondly, alcohol as stated is significantly associated with ESCC. While an alcohol related signature is not reported in COSMIC it was reported by Li et al1 in their recent study of 549 exomes of ESCC. Munmee et al should clearly state how faithfully the alcohol signatures identified in both studies concur in their cosine similarity and 96-trinucletide context profile. Alternative signature extractions such as SigProfiler methods if necessary could aid in this.

2) Chang et el2 have previously sequenced 94 ESCC cases using Whole genome sequencing and in the Chinese population and identified a total of ~610,000 SNVs in their cohort with a mutational burden of 2.1 mutations/Mb. This is in clear contrast to the ~104,000 mutations identified in the 10 samples. Munmee et al should clarify if these differences are due to difference in disease aetiology or technical differences such as depth of coverage or purity.

3) While the sample size for driver detection using mutations is small for methods like MutsigCV, methods such as dNdsCV should be able to delineate driver genes from those mutated by chance alone.

4) Chang and Li et al in their previous publications showed NOTCH1, KMT2D, PIK3CA as the most commonly mutated genes apart from TP53 and FAT1 with frequencies of each gene being mutated being > 10% frequency. However, these mutations are not present in any of the samples in this cohort. Can these differences in mutated genes be attributed to differences in the Chinese and Japanese population at disease presentation? IF mutations are present, showing these driver mutations in Figure 1 is more relevant that showing genes mutated by chance and not conferring the tumors cells any growth advantage.

5) Cheng et al3 in another WGS study of ESCC have reported recurrent SVs like in this cohort have reported recurrent inactivation of CDKN2A cohort and activations of FGFR1 and CCND1 in their cohort. They report the mechanism of activation of CCND1 is through Breakage-Fusion-Bridge (BFB) and this is clearly seen the WGS data through the stepwise increment in the coverage. In their study, they show chromothripsis as common method of activation of FGFR1. Does a similar pattern of activation found in this cohort?

6) Chromothripsis, Katetgis, BFB and double-minutes are identified in more than half of the samples in Cheng et al. Have these mechanisms of activation and inactivation’s been identified in your cohort also?

Minor revisions
1) Gene such as TTN are large and frequently mutated without any known biological relevance due to their size and should be removed from Figure 1A. Driver detection algorithms account for size and such genes are usually excluded.
2) Samples are referred to as OK007, OK008, etc..(line 208). While Figure 3A has labels like ESCC_007. The label names should be kept consistent.

References
1 Li, X. et al. A mutational signature associated with alcohol consumption and prognostically significantly mutated driver genes in esophageal squamous cell carcinoma. Annals of Oncology 29, 938-944 (2018).
2 Chang, J. et al. Genomic analysis of oesophageal squamous-cell carcinoma identifies alcohol drinking-related mutation signature and genomic alterations. Nature communications 8, 15290 (2017).
3 Cheng, C. et al. Whole-genome sequencing reveals diverse models of structural variations in esophageal squamous cell carcinoma. The American Journal of Human Genetics 98, 256-274 (2016).

Annotated reviews are not available for download in order to protect the identity of reviewers who chose to remain anonymous.

---

## Round 0.2 · accepted · Accept

Thank you to submit this interesting paper for PeerJ to provide a unique dataset in the Japanese population and conduct a commendable job in identifying the key driver Structural Variants and Copy Number events.

·

Basic reporting

no comment

Experimental design

no comment

Validity of the findings

no comment

Additional comments

The revised manuscript corrected for most of the issues raised in the review. In particular, some methodological details and missing references were added. In addition, the authors revised several statements in the text that were either vague or unfounded based on their analysis. The issue of reproducibility remains however. While not strictly required by the journal, the authors can still reaffirm their commitment to reproducible analysis by providing 1) links or accession numbers to the deposited data 2) links to the code that was used to perform the analysis and generate the figures/tables.